# Mazes to Study the Effects of Spatial Complexity, Predation and Population Density on Mate Finding

**DOI:** 10.3390/insects11040256

**Published:** 2020-04-20

**Authors:** Lloyd D. Stringer, Nicola J. Sullivan, Robyn White, Alfredo Jiménez-Pérez, Jess Furlong, John M. Kean, Jacqueline R. Beggs, David Maxwell Suckling

**Affiliations:** 1The New Zealand Institute for Plant and Food Research Limited, Christchurch PB 4704, New Zealand; Nicola.Sullivan@plantandfood.co.nz (N.J.S.); Robyn.White@plantandfood.co.nz (R.W.); Jess4long@gmail.com (J.F.); Max.Suckling@plantandfood.co.nz (D.M.S.); 2Better Border Biosecurity (B3), Lincoln 7608, New Zealand; john.kean@agresearch.co.nz; 3Centre for Biodiversity and Biosecurity, School of Biological Sciences, University of Auckland, PB 92019, Auckland 1142, New Zealand; j.beggs@auckland.ac.nz; 4Centro de Desarrollo de Productos Bióticos, Instituto Politécnico Nacional, Yautepec 62731, Mexico; aljimenez@ipn.mx; 5AgResearch, PB 3123 Hamilton 3240, New Zealand

**Keywords:** Allee effects, *Chelifer cancroides*, *Drosophila*, maze, population dynamics, spatial, 3D

## Abstract

The difficulty to locate mates and overcome predation can hamper species establishment and population maintenance. The effects of sparseness between individuals or the effect of predators on the probability of population growth can be difficult to measure experimentally. For testing hypotheses about population density and predation, we contend that habitat complexity can be simulated using insect mazes of varying mathematical difficulty. To demonstrate the concept, we investigated whether the use of 3D printed mazes of varying complexity could be used to increase spatial separation between sexes of *Drosophila simulans*, and whether the presence of a generalist predator hampered mate-finding. We then examined how increasing *D. simulans* population density might overcome the artificially created effects of increasing the distance between mates and having a predator present. As expected, there was an increase in time taken to find a mate and a lower incidence of mating as habitat complexity increased. Increasing the density of flies reduced the searching time and increased mating success, and overcame the effect of the predator in the maze. Printable 3D mazes offer the opportunity to quickly assess the effects of spatial separation on insect population growth in the laboratory, without the need for large enclosed spaces. Mazes could be scaled up for larger insects and can be used for other applications such as learning.

## 1. Introduction

Population growth rate can be affected by positive density dependence, which is also known as inverse density dependence or the Allee effect at low population density, whereby there is a positive relationship between population density and growth rate. [1,2,3]. Allee effects may cause longer lag times for population growth, and subsequently slow the growth and spread rate and lower the likelihood of persistence of both invasive populations and populations under conservation management [4,5]. Although an Allee threshold suggests a definitive value, in reality, the threshold is probabilistic, with a higher probability of population growth above the threshold and probable reduction to extinction below. Moreover, the thresholds may vary depending on spatial and temporal ecosystem factors [6]. Allee effects may be due to several factors, including reduced mating, which may become problematic for low density populations [7]. Positive density dependence may be exploited for population management where not all of the population necessarily needs to be killed to achieve eradication [8], or to ensure that population density is sufficient for ongoing continuation of a population [9,10].

To date, Allee effects in insects have been quantified mostly from large-scale multi-year population studies. For example, reduced reproductive success and increased predation of low-density Monarch butterfly *Danaus plexippus* populations [9,11]. Another study has estimated a trap-catch-based Allee threshold for gypsy moth *Lymantria dispar dispar* from multi-year trapping data from multiple trap transects placed through the invasion front of the moth in North America [8,12]. Such trials use extensive field datasets and analyses to understand what factors promote or have the greatest effect on population persistence.

Trying to assess the effect of spatial separation on mate finding and mating success is difficult in the laboratory as space is limited. Study systems must be large enough to achieve sufficient distance between individuals, but small enough that the individuals can still be tracked by the observers. The advent of 3D printers has enabled the production of complex shapes, such as mazes, to be readily created. Mazes of known mathematical complexity can be readily used for hypothesis testing [13]. To test the concept, we investigated whether mazes of varying complexity could be used to increase spatial separation to investigate mate-finding Allee effects in the laboratory using *Drosophila simulans*.

Mazes have often been used to investigate learning in organisms (e.g., heat mazes in *Drosophila melanogaster* [14] and water mazes for rats [15]). However, to the best of our knowledge, mazes have not been used to modify spatial distance between individuals. We hypothesised that mazes of increasing complexity would increase the time taken for one male and one female *Drosophila simulans* placed at opposite ends of the maze to find each other, artificially increasing the Allee threshold. This is due to an increase in the length of the route needed to be taken through the maze, reducing the probability of mating occurring within a fixed time. Furthermore, we hypothesised that the presence of a generalist pseudoscorpion predator (*Chelifer cancroides*) in the maze would increase the time taken for pairs of flies to find each other, again increasing the Allee threshold. However, we predicted that the presence of multiple pairs of flies in the maze, increasing fly density, would overcome the negative effects of the spatial separation and predation on mate finding and frequency of mating.

## 2. Materials and Methods

*D. simulans* colonies were maintained on a diet of potato flakes (128 g), casein (28 g), castor sugar (20 g), baking yeast (8 g), methyl paraben (0.6 g), ascorbic acid (1.3 g) and benzoic acid (0.6 g) at 25 °C in ambient day length. Twenty millilitres of the diet/oviposition material was added to a 120 mL circular vial (110 mm × 43 mm i.d.), and 20 mL of water was added to hydrate the diet/oviposition material. As *D. simulans* pupate out of the diet on the edge of the vial, the vials were lined with waxed baking paper so that pupae could easily be removed. To facilitate pupal removal, the baking paper was sprayed lightly with water and each pupa was placed into a polycarbonate test tube (75 mm long, 10 mm i.d.) stoppered with a water-moistened cotton wick. Pupae-filled tubes were kept in a sealed bag to maintain high levels of humidity. No food was added to these tubes. Tubes were checked once daily for eclosion. On eclosion, the adults were sexed and 30% (by volume) sugar water was added to the cotton wick. Adult flies were used once when they were between 48 and 72 h old, when the majority of *Drosophila spp.* are sexually receptive [16].

### 2.1. Chelifer cancroides Rearing

The *Chelifer cancroides* colony of many hundred individuals was maintained on a diet of *D. simulans* and cultured following Read et al. [17]. The *D. simulans* were maintained as above. When *D. simulans* started to pupate in the vials, an additional 20 mL of water was added and the larvae/diet solution was gently mixed so that it could be poured into the *C. cancroides* colony weekly. The *C. cancroides* were observed feeding on adult flies but may have also fed on the larvae or the diet itself. For the trials using *C. cancroides*, five sub-colonies of six males and six females were developed from the main colony; one colony for each day of the working week. Sub-colonies were fed a diet of ~20 mixed sex adult *D. simulans* once a week immediately after participating in trials. The following day all dead and live flies were removed from the colony so that the colony was starved for six days prior to use. They had access to water ad libitum and tissue paper to hide in. The colonies were maintained at 25 ± 2 °C in ambient day length.

### 2.2. Maze Creation Overview

Two-dimensional mazes with variable complexity were generated on the website, www.mazegenerator.se. Circular mazes were set to have an “outer diameter” of 20 cells (90 mm). The “internal diameter” setting varied with the desired complexity. Four maze complexities were generated: simple, with an external and internal diameter of 20 cells (only an outer wall, similar to a Petri dish); easy, with an external diameter of 20 cells, internal diameter of 14 cells (three cells of maze); medium, with an external diameter of 20 cells, internal diameter of eight cells (six cells of maze); and hard, with an external diameter of 20 cells and an internal diameter of four cells (eight cells of maze). Adjacent walls (cells) were 3 mm apart and the height of the maze was 5 mm, and a Perspex lid was added to the top of the maze to prevent fly escape and facilitate observations. Two additional parameter values—elitism and river—were set to zero in the maze generator programme. The elitist value controls the solution length of the maze. The non-elitist (elitism = 0) parameter weights the maze generator programme to generate a maze that has a solution path that goes through a large portion of the maze, conversely an elitist maze (elitism = 100) has a short solution length from the edge to the centre of the maze. The river value controlled the length of dead-ends. A high river value (river = 100) weights the programme to generate a maze with few but long corridors that lead to a dead end, whereas a low river value of 0, as used here, has many short corridors that lead to dead ends (Figure 1). Technical details are provided in Appendix A.

### 2.3. Spatial Complexity Trial

A single pair of flies was tested to determine whether the time to find a mate and the probability of mating changed with increasing spatial separation, caused by the escalating complexity of the maze. In these microcosms, flies were given one hour to find each other and mate. Males were released through the hole in the centre of the maze and females through the side entrance generated by the maze generator. The entrances were then closed with masking tape. The four complexity levels (simple, easy, medium and hard) were tested simultaneously and replicated 30 times. Time taken for mate location was recorded, as well as whether the pair mated. This was assessed visually and any pairs mating were recorded as successful if the pair remained in copula (minimum 14 and maximum 26 min in copula recorded), without the female trying to avoid the male.

### 2.4. Fly Density

We tested whether there was an increase in the probability of mating and a reduction in the time for pairs to find each other when there was a higher density of flies in the maze. One, three and five pairs of flies were tested in the “easy” solution maze. In addition, to determine whether there would be any effect of a predator on mate finding and mating, we added a 6-day starved male *Chelifer cancroides*. The predator was added to the halfway point between both the male and female maze entry points immediately prior to the addition of the flies. Each replicate comprised one of each of the three fly density levels with and without a predator (six mazes total per replicate). After the predator was added, male flies were placed in the centre of the maze and held there using a thin (1 mm) gate inserted through the Perspex lid until all female flies were added to the maze through the side entrance. When all flies were present in the maze, the males were allowed to leave the centre and the trial was started and ran for a maximum of one hour as above. All six treatments were run simultaneously for each replicate (n = 30 replicates). Time for the first pair to find each other was recorded, as well as any predation, sex of the predated fly and the number of pairs mating following the criteria above.

The minimum distance that the flies had to travel to meet each other was investigated by using the tracking software MaxTRAQ v1.91 (Innovision Systems, Lapeer, MI, USA) (Figure 2). The distance to solve the maze from edge to centre was measured and the distance was divided by two assuming that both insects would move towards each other at the same rate.

### 2.5. Statistical Analysis

To model the effect of spatial complexity on maze solution length and percentage of single-pair fly trials where flies found each other, we performed a linear regression using Origin v. 8.5 (OriginLab Corporation, Northampton, MA, USA). To analyse the effect of the number of pairs of flies and the presence of a predator on time to find a mate, and the probability of mating occurring, we used a Generalized Linear Model (GLM) with a Poisson distribution logarithm-link function, and a GLM with a binomial distribution logit-link function, respectively. The GLM analyses were performed in GenStat v. 17 (VSN International Ltd., Hemel Hempstead, UK).

## 3. Results

The flies failed to perform in the way we anticipated as single pairs, with only three of the 30 pairs of flies mating in the simple maze and one pair of flies mating in the medium complexity maze. No flies mated in the easy or hard complexities in the one hour allotted. However, as we predicted, fly mating rates increased with increasing fly density.

### 3.1. Spatial Complexity

The maze experiments showed a positive linear relationship between the complexity of the maze and the solution length (R^2^ = 0.984). The minimum distance flies needed to travel to meet each other increased by 7-, 10- and 15-fold for flies in the easy, medium and hard mazes, respectively, compared to the flies in the simple maze.

The maze complexity also affected the percentage of flies that found a mate in the 1 h period (R^2^ = 0.957) in the single pair treatments (Figure 3).

The probability that a pair of flies would find each other was significantly different between the treatments as assessed by a Generalized Linear Model (GLM), binominal distribution logit-link (deviance = 12.6, 141.2; d.f. = 1; *p* < 0.001) for the main interaction for the complexity of the maze, explaining approximately 91% of deviance observed. Of the flies that found each other, the time taken to do so also increased with an increase in the complexity of the maze; GLM-Poisson log-link (deviance = 173, 1403; d.f. = 1; *p* < 0.001) main interaction of the complexity on time to find each other explaining 87% of the deviance. However, this difference was due to the difference between the simple maze and the three other complexities only; *p* = 0.115 for the comparison between the time taken in the easy, medium and hard complexities (Figure 4).

### 3.2. Fly Density

The average time for one fly of each sex to find the other varied with the number of pairs that were in the maze and whether there was a predator present (Table 1). The time for initial mate finding was faster in the multiple pair fly treatments than for the single pair of flies. The presence of a predator in the maze increased the time it took for all flies to find each other (Table 2), with a total of 18 male and 15 female flies killed by the predators during the experiment (https://youtu.be/BEaRsO0h2v0).

Despite increasing the time it took for individuals to find each other in the maze, the presence of a predator did not reduce the probability that at least one pair would mate. There was no effect of the interaction term of the predator presence and number of pairs mating, so this was removed from the analysis and main effects only are shown (Table 3). The biggest effect on mating was the number of pairs present in the maze, with an increase in the percentage of flies mating as the density of flies in the maze increased.

## 4. Discussion

Mazes of increasing complexity increased the spatial distance between individual *D. simulans*, providing a baseline for testing mate-finding effects in this system. As complexity (solution length) increased, the initial distance between flies increased, requiring individual flies to spend a prolonged amount of their finite time searching for each other. By reducing encounter rates in the limited area of the 90 mm diameter arena via increased spatial separation, we expected to manipulate the mate-finding Allee threshold relative to the Petri dish style simple maze.

The probability that flies would find each other within the one hour allowed for the flies to navigate the maze decreased with increasing complexity, with all flies finding each other when there was only an outer wall present (similar to a Petri dish). Over half of the flies did not find each other in the most complex maze offered; a maze that had a route that was 15-fold longer than the simple Petri dish style maze.

At the individual fly level and replicate “population” level, the increase in fly density was able to overcome the spatial separation effects and the effect of the single predator in the system. The addition of multiple flies increased fly density in the mazes, the percentage of females mating and percentage of any mating event at all occurring per replicate in both the predator present and predator absent trials. In the single pair treatments, only 73% of trials had the potential to have a mating event occur because the predator killed flies in eight of the 30 trials. Even though a greater number of flies overall were killed in the multiple pair fly trials, flies were killed at a similar rate between the different fly density treatments, and all multiple pair fly trials had flies of each sex remaining at the end of the trial. We recognize that by printing mazes that had open corridors of approximately 3mm width with a height of 5 mm, there was limited ability for the Drosophila to use flight, a prime mode of their locomotion for predator avoidance. However, as we were investigating the role of habitat complexity and fly density on mate finding ability and subsequent copulation, their reliance on short hops and running here was considered sufficient for these treatments. The increase in the number of flies in the mazes increased the density of the flies, thus increased the probability of mating occurring, but it may have also promoted an increased rate of dispersal from the respective release area for the males and due to perceived competition [18]. We did not measure parameters such as walking speed or area of maze traversed to estimate dispersal rate to determine whether this phenomenon could have been a factor that influenced time to mate finding rates in our trials.

In the single pair trials, the presence of the predator increased the time that was taken for the pair of flies to meet. This reduced the probability that mating would occur partly through direct predation and because of the increased searching time required in the presence of the predator potentially altering fly behaviour in order to avoid the predator [19]. When the predator was present only one of the single pair populations or 3% of the replicates had a mating event, whereas 43% and 46% of the three and five pair populations, respectively, were successful at mating. Under the trial conditions of only one hour to perform mating, the higher fly density populations would be more likely to persist than the single pair populations. This increase in fly density led to a greater chance of finding and successful mating by pairs and a reduced likelihood of attack potentially because of a Type II functional response with satiation of the natural enemy [20]. A difference in prey handling time or multiple predators would probably shift the fly density requirement up to maintain a population as more prey would probably be removed from the population potentially leading to a decline in the population growth rates [21]. Further, non-mortal sources of reduced mating can be caused through the perception of a predator via senses such as smell and vibration that induce avoidance responses in *Drosophila*, which can result in behaviours that depress the per capita population growth rate [19]. The presence of the predator used here, *C. cancroides*, in the maze did increase the time taken for the first pairs of flies to find each other, but not appear to have an effect on the probability of fly mating, unlike the increase in fly density that lead to 6 or 8 times the probability of any mating occurring in the three and five pair trials respectively, than was observed in the single pair trials.

An increase in spatial separation between the sexes increased the time taken for individual flies of the opposite sex to find each other from simple to easy, but not for the easy–hard complexities. The initial change in spatial separation from the simple to easy maze was 13 cm distance or an increase ×6.9 the distance. Further, for the males to escape the centre of the maze they were required to find the 3 mm gap in the inner wall to enter the walled section of the 3D-printed maze. The increase in distance from easy to hard complexity was 18.1 cm or ×2.2 the distance. This distance may have been insufficient to observe noticeable changes in the time taken for the flies to navigate the maze. Further, once flies were in the walled section of the maze they may have quickly negotiated the corridors. Unfortunately, we did not quantify arrestment behaviours of the flies related to the sections of maze they were in.

Kramer et al. [22] used a meta-analysis approach on data from natural populations and concluded that the most common sources of Allee effects were from mate limitation, reduction in cooperative defence, reduced predator satiation, cooperative feeding or over-dispersal and change in habitat, concurring with earlier authors [1,5,23,24]. The maze used here offers a simple way to assess the fundamental effects of mate limitation and predator satiation on population growth. Mazes could be used for a large number of other behavioural applications including traditional aspects of organism learning. The 3D printed mazes can also work for larger organisms; a pilot study assessed a larger version of the maze for Queensland fruit fly, *Bactrocera tryoni*. Here, we have observed and tested the role of mate limitation through varying fly density and predator satiation or predator-induced avoidance behaviours exhibited by the flies in a novel use of 3D printed mazes of varying complexity.

## 5. Conclusions

Flies were able to successfully navigate and mate during their natural dusk mating period. By overhead recording of the maze trials, any assessment of behavioural responses to non-visual stimuli can be assessed with image analysis. Natural dispersal was limited in the maze system, as the flies were forced to walk rather than fly. Our next step is to determine the correlation between maze complexity and real world distance, in order to model real-world spatial separation effects from laboratory-derived datasets.

## Figures and Tables

**Figure 1 insects-11-00256-f001:**
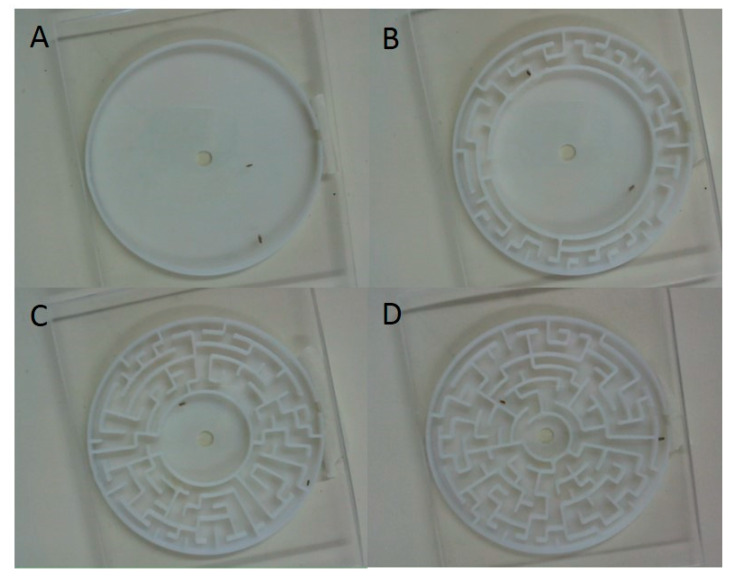
Four mazes were used with different complexities (**A**—simple; **B**—easy; **C**—medium; **D**—hard) to test factors affecting mate finding in *D. simulans*. A Perspex lid was temporarily fixed to the top of the maze to allow for observation of the fly interactions.

**Figure 2 insects-11-00256-f002:**
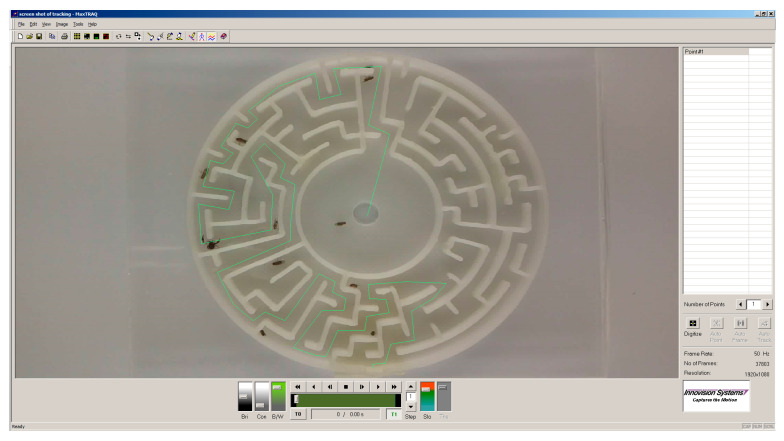
Illustration of the MaxTRAQ movement tracking software. A maze containing *D. simulans* and a male *C. cancroides* is shown. The shortest possible solution has been digitised manually (green line) to measure the distance.

**Figure 3 insects-11-00256-f003:**
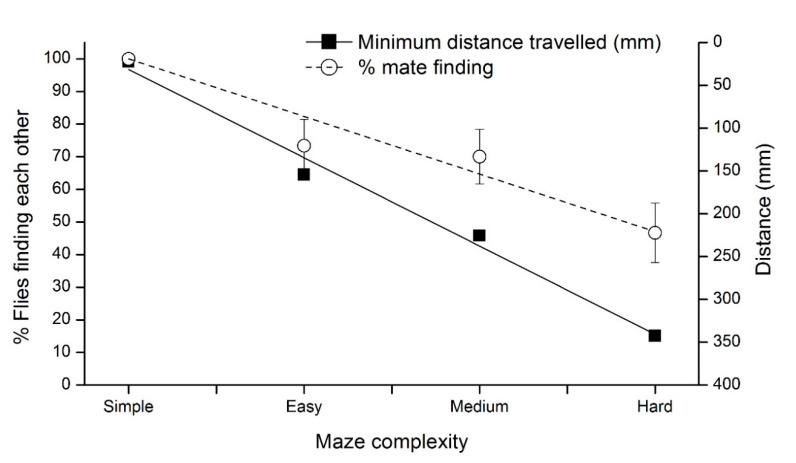
Relationship between the complexity of the maze and the minimum walking distance required for a pair of *D. simulans* to encounter each other, and the percentage of trials where pairs of flies found each other after starting at opposite ends of the maze.

**Figure 4 insects-11-00256-f004:**
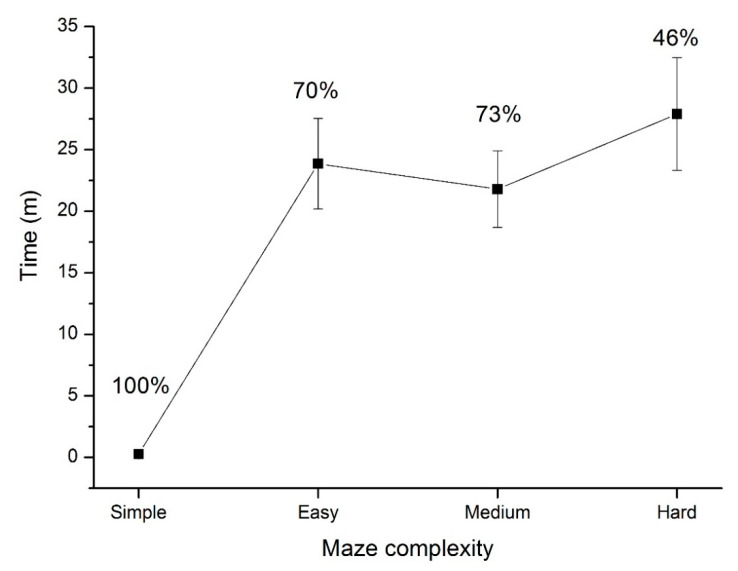
In the trials where a single pair of *D. simulans* found each other, the mean (+SEM) number of minutes taken for the pairs to find each other in mazes of different complexity is displayed. The percentage of replicates that flies found each other within the 1 h time limit is shown for each maze complexity.

**Table 1 insects-11-00256-t001:** Time for the first female and male *D. simulans* to find each other in a maze with or without a pseudoscorpion predator (*C. cancroides*) present and the percentage of flies finding each other within a treatment in trials with varying fly density.

No. of Pairs	Predator Present	Mean Initial Discovery Time	% of Trials with Successful Mate Finding	% of Trials with Mating Occurring	Percentage of Flies Mating	Number of Flies Killed (In N Replicates)
1	Yes	25m15s	53%	3%	3%	8 (8)
1	No	17m10s	76%	10%	10%	-
3	Yes	16m35s	96%	43%	15%	12 (11)
3	No	16m31s	96%	40%	20%	-
5	Yes	12m40s	100%	46%	13%	13 (10)
5	No	7m28s	100%	60%	25%	-

**Table 2 insects-11-00256-t002:** GLM Poisson log-link regression of time taken for a pair of *D. simulans* to find each other (first pair only), with or without a *C. cancroides* (predator) present. The reference parameters are one pair of flies and no predator present. The antilog can be considered as the multiple by which the change has occurred compared to the expected result for the reference parameters (1 pair and no predator).

Parameter	Estimate (s.e.)	t	*p*	Antilog of Estimate
3 pairs	−0.2372 (0.0480)	−4.95	<0.001	0.7888
5 pairs	−0.7345 (0.0540)	−13.61	<0.001	0.4797
Predator present	0.2651 (0.0413)	6.41	<0.001	1.303

**Table 3 insects-11-00256-t003:** GLM binomial logit-link regression of the probability that mating would occur at all in trials where different numbers of pairs of *D. simulans* were in the presence of a predator (*C. cancroides*) or not. The reference parameters were one pair of flies and no predator present. The antilog can be considered as the multiple by which the change has occurred compared to the expected result for the reference parameters (one pair and no predator).

Parameter	Estimate (s.e.)	t	*p*	Antilog of Estimate
3 pairs	1.844 (0.538)	3.43	<0.001	6.324
5 pairs	2.095 (0.530)	3.96	<0.001	8.127
Predator present	−0.167 (0.259)	−0.64	0.519	0.8463

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
