# Peer review of "Mazes to Study the Effects of Spatial Complexity, Predation and Population Density on Mate Finding"

_insects, 2020, doi:10.3390/insects11040256_

Round 1

Reviewer 1 Report

The mazes are designed well but this brings to mind the question of flying in mating and avoidance behavior - is it necessary?

Would the same results occur in a 3D maze? The Allee effect should be highlighted a bit more and connected to the hypothesis.

The fly density assay is well designed, and he result that the number of pairs present affects the mating rate is interesting.

It seems that the Chelifer doesn't prefer one sex of fly over the other, but I think that is an avenue worth investigation.

Reviewer 2 Report

The manuscript gives a good contribution to spatial effects on predators using mazes. However, I have concerns about some statistical methods and values of the paper that I believe need to be addressed in order to improve its clarity. Their approach is interesting but it has some flaws that make this version unacceptable for publication. Provided they conduct changes to the manuscript, I believe this paper could be of interest to the interested reader on population insect density.
Please see my specific comments below:
L.41: Place “,” after reality
L.51: … gypsy moth, Lymantria dispar from…
L.65 and L.69: we hypothetized…
L.75 Colonies…Which species?
L.77-78: Change ml by mL
L.79: Since D. simulans…
L.86: majority of D. simulans are…
L.90: Change ml by mL
L.150-151: Drosophila stimulans in italic
L.151: …digitized…
L.74-148: In material and methods, provide information about different statistical methods used in each experiment.
L.170: Delete “significantly”
L.172 and 176: Provide the χ2 value
L.183-189: Again, there is no statistical data information.
Ls.268-270: These sentence is no conclusive, place by discussion or delete.

Round 2

Reviewer 2 Report

The manuscript “Mazes to study the effects of spatial complexity, predation and population density on mate finding” has been improved and all my questions were taken into account. I recommend the publication in “Insects”.

Author Response

Dear Reviewer,

I appreciate the time you have taken to consider our original submission and our revisions based on your very helpful suggestions.

Sincerely,

Lloyd Stringer